IOCBIO Sparks detection and analysis software

Laasmaa Martin
Karro Niina
Birkedal Rikke
http://orcid.org/0000-0002-6459-0391 Vendelin Marko markov@sysbio.ioc.ee
Laboratory of Systems Biology, Department of Cybernetics, School of Science, Tallinn University of Technology , Tallinn , Estonia
Gollo Leonardo
Electronic publication date: 2019 Mar 29
Publication date: 2019
Volume: 7
Electronic Location ID: e6652
Received 2019 Jan 7; Accepted 2019 Feb 19
Copyright: © 2019 Laasmaa et al.
Copyright year: 2019
Copyright holder: Laasmaa et al.
License: This is an open access article distributed under the terms of the Creative Commons Attribution License, which permits unrestricted use, distribution, reproduction and adaptation in any medium and for any purpose provided that it is properly attributed. For attribution, the original author(s), title, publication source (PeerJ) and either DOI or URL of the article must be cited.
License URL: https://creativecommons.org/licenses/by/4.0/

Keywords: Analysis software, Open source, Cardiomyocyte, Calcium spark, Mouse, Confocal microscopy

Funding: Estonian Research Council IUT33-7 This work was supported by the Estonian Research Council [IUT33-7]. The funders had no role in study design, data collection and analysis, decision to publish, or preparation of the manuscript.

==============================
Analysis of calcium sparks in cardiomyocytes can provide valuable information about functional changes of calcium handling in health and disease. As a part of the calcium sparks analysis, sparks detection and characterization is necessary. Here, we describe a new open-source platform for automatic calcium sparks detection from line scan confocal images. The developed software is tailored for detecting only calcium sparks, allowing us to design a graphical user interface specifically for this task. The software enables detecting sparks automatically as well as adding, removing, or adjusting regions of interest marking each spark. The results of the analysis are stored in an SQL database, allowing simple integration with statistical tools. We have analyzed the performance of the algorithm using a large set of synthetic images with varying spark sizes and noise levels and also compared the analysis results with results obtained by software established in the field. The use of our software is illustrated by an analysis of the effect of isoprenaline (ISO) on spark frequency, amplitude, and spatial and temporal characteristics. For that, cardiomyocytes from C57BL/6 mice were used. We demonstrated an increase in spark frequency, tendency of having larger spark amplitudes, sparks with a longer duration, and occurrence of multiple sparks from the same site in the presence of ISO. We also show that the duration and the width of sparks with the same amplitude were similar in the absence and presence of ISO. The software was released as an open source repository and is available for free use and collaborative development.

Introduction

During cardiac excitation-contraction (EC) coupling, a rapid change in the cell’s membrane potential initiates a transient change in intracellular calcium concentration. This change is mainly caused by the opening of L-type Ca2+ channels in the sarcolemma allowing Ca2+ to enter the cell, which in turn triggers Ca2+ release from the sarcoplasmic reticulum (SR) by the opening of ryanodine receptors (RyRs) in the SR (Bers, 2002). Ca2+ release from the RyRs can also occur spontaneously in an unstimulated cell (Cheng & Lederer, 2008). These elementary Ca2+ release events in a cell can be visualized with a confocal microscope by recording changes in fluorescence of a Ca2+ sensitive dye (Ca2+ sparks), which occur as a brief, localized increase in the recorded signal (Cheng, Lederer & Cannell, 1993). It has been shown by many researches that Ca2+ sparks properties differ in health and disease, reviewed in (Hoang-Trong, Ullah & Jafri, 2015). Therefore, studying different spark characteristics like size, shape, dynamics, and frequency of Ca2+ sparks can provide valuable information about functional changes occurring in cardiomyocytes during heart failure. For example, a study by Louch et al. (2013) on a congestive heart failure mouse model suggests that slowed and de-synchronized Ca2+ release during the action potential is caused by a population of Ca2+ sparks with slow kinetics.

It is estimated that at cell rest there are roughly 100 Ca2+ sparks per second per cell (Cheng, Lederer & Cannell, 1993). Hence, having a specialized software for spark detection and characterization is necessary (Cheng et al., 1999). While there are several programs available for detection and analysis of calcium sparks, there is a lack of open source automated software tailored for detection of sparks. Namely, either license and source code are not available, as in the popular ImageJ plugin (Picht et al., 2007), no source code available but there is a license to use it (Steele & Steele, 2014), or, even with the source code available, no license is given (Davoodi et al., 2017). Some of the sparks detection software has been written using closed source analysis environments with limited availability, such as IDL or MATLAB (Davoodi et al., 2017; Cheng et al., 1999; Kong, Soeller & Cannell, 2008), making them less attractive for further development in open source manner.

The aim of this work was to develop an open-source platform for automatic calcium sparks detection from line scan confocal images using open source components. While developing the software, we also targeted several specific requirements. First, since the software was developed for detecting calcium sparks only, we could design a graphical user interface (GUI) specifically for this task (see Fig. 1 for a screenshot). Second, while providing automatic sparks detection, we wanted to make it possible for the user to either add, remove, or adjust region of interest marking each spark. Lastly, the output is saved to an SQL database for inter-operability with third-party applications.

Figure 1 Overview of the graphical user interface.

See text in ‘Results’ for description.

Methods

Algorithm for sparks detection

The detection algorithm is based on thresholds approach by Cheng et al. (1999) and Picht et al. (2007). Compared to earlier algorithms, there are some modifications in the determination of the image background and normalization of the image, as described below. While the default analysis parameters were taken as in Picht et al. (2007), users can now adjust them and define new defaults in the implemented program. The flowchart of the algorithm is shown in Fig. 2.

Figure 2 Schema of the spark detection algorithm.

See text for description.

As the first stage of the spark detection, the background image F0 was estimated. It is started on the basis of the measured images with the dark current of the detector subtracted, as specified by the user. Below, the measured image with the subtracted dark current is referred to as “image.” F0 was obtained as a spline fit to the image with the regions with autodetected sparks excluded from the fit. The process is iterative, as shown in the top left part of Fig. 2.

During each iteration, intensity changes in the regions without sparks were fitted by a third order one-dimensional spline with the user-prescribed knots placement, 5 s apart by default, one spline fit per each location in the cell (x in xt images). Depending on the studied cells, expected spark sizes, the distance between knots can be changed by the user. Note that the distance between knots should be considerably larger than the expected spark duration to avoid interpolation of the background influencing the spark parameters.

For spark detection, the image was corrected by subtracting F0 and dividing it by the square root of the background intensity. The division was performed to ensure even distribution in x of the corrected image noise, as expected for Poisson processes (see Fig. 3). This correction is similar to the variance stabilization in (Bankhead et al., 2011). After correction, the image was smoothed by median and boxcar filters. On such corrected and smoothed image, sparks were detected as joint regions with pixel intensity higher than SparkIntensity × SD, with at least one pixel in the region with intensity higher than SparkPeakIntensity × SD, where SD is the standard deviation of the corrected smoothed image and SparkIntensity, SparkPeakIntensity had values of 2.0 and 3.8, as default. In agreement with the algorithm by Cheng et al. (1999), binary image of intensity > SparkPeakIntensity × SD condition was processed using median filter to avoid false positive detection of sparks induced by a single high-value pixel. In contrast to Cheng et al. (1999), we did not processed intensity > SparkIntensity × SD binary image with the median filter, but allowed user to specify a minimal size of a spark in pixels, with the spark region defined as a connected area where condition intensity > SparkIntensity × SD was valid. In the following analysis, minimal spark region of 40 pixels was imposed (can be specified by the user of the program). With the spark regions detected, the image without them was used to make the next F0 estimation. The iteration continued until convergence or for a given number of iterations.

Figure 3 Spatial distribution of the image background noise without (blue) and with (red) the correction of the image.

Notice how accounting for the nature of Poisson process of image detection in confocal microscope evens the noise distribution.

After estimation of the background, sparks are automatically detected using the smoothed image with the corrected mean and SD using the criteria specified above. The automatic detection determines the regions of interest (ROI) and the spark peak location, one per spark. Fluorescence changes, as on the image with the subtracted background, within each ROI, are analyzed by fitting temporal and spatial changes in fluorescence for data crossing the peak by a spline. Each automatically detected spark is assumed to be acceptable by default. The program, however, allows the user to go through all sparks and visually inspect each ROI as well as the quality of the fluorescence changes fits. In some cases, spline parameters, ROI location, and dimensions have to be changed to obtain a better fit. In addition, the user may deem necessary to delete an automatically detected spark. Such a review of the fit is an iterative process for each spark until the user is satisfied with the fit and detected sparks.

Storage of analyzed data

The analysis results are stored in an SQL database. The software supports storing data in a central (PostgreSQL) or a local database (SQLite). From the local database, users can either transfer the data into some central database, analyze using local database tools, or export in spreadsheet form.

The database schema was designed for simple integration into larger laboratory databases by allowing to link sparks recordings with the cardiomyocyte preparation, animals used, and other data. In our practice, we link the experiments with the description of cardiomyocyte isolation data allowing us to study sparks properties in relation to the animal properties. Each spark is marked by a unique ID, allowing to store the data from all experiments in the same database.

Implementation and availability

The source code of the developed program is released under GPLv3 license and available at GitHub (https://gitlab.com/iocbio/sparks). The program is written in Python 3, using, among others, the following packages: numpy (Oliphant, 2006), Scipy (Jones, Oliphant & Peterson, 2001), scikit-image (Walt et al., 2014), PyQt5, PyQtGraph. Software installation is covered on the homepage with the instructions for installation on several platforms.

Evaluation of algorithm performance

To evaluate algorithm performance, we generated synthetic data, which was then analyzed the same way as the experimental data.

Synthetic data. For an artificial spark, the following formula was used: S(x,t)=R⋅F0⋅2−[(xξ)2+(tτ(t))2]and τ (t)={τrif t<0τdotherwise,

where x and t are spatial coordinate and time, respectively, R is relative amplitude, ξ is half of the spark’s full width at half maximum (FWHM), τr and τd are the time constants (τr + τd is equal to spark’s full duration at half maximum (FDHM)). For all sparks, spatial and time constants were the same: ξ = 1.5 μm, τr = 7.0 ms, and τd = 18.0 ms. For the amplitude R, we used values from range 0.05–0.8 with the step of 0.05 and, additionally, values 1.0, 1.25, 1.5, 2.0 (in total 20 different values).

The synthetic experiments covered ∼70 μm in space (512 pixels, 0.14 μm pixel size) and were performed for ∼60 s (37,000 lines, 1.53 ms apart) with the spark frequency of 1.5 sparks/(s · 100 μm) resulting 61 sparks per a synthetic experiment. Experiments were performed at different F0 levels (1, 2, 3, 4, 6, 9, 12, 16). In the experiments, sparks were added to an experiment at random locations. Due to the quantum nature of light, the number of detected photons in a confocal microscope is a Poisson process (Laasmaa, Vendelin & Peterson, 2011), and therefore the synthetic experiment was degraded with Poisson noise.

For 8-bit synthetic image preparation, the synthetic data was multiplied by a gain factor γ=255 (α+5α),where α=F0(1+maxR),

which scaled the data to fill large fraction of values from 0 to 255. Next, after scaling with the gain factor, data was converted to 8-bit integer representation, as in the confocal that was used for the experiments.

In the analysis, we used images with the different signal-to-noise ratio (SNR). Here, SNR was defined as SNR of the background and calculated as a mean value of the background divided by the standard deviation of the background.

Synthetic data analysis

For each synthetic image, we stored spark locations in a database as a “ground truth” reference. After that, the same algorithm, as for real experiments, was applied to detect sparks from the synthetic data. From the comparison of the spark locations ground truth with the detected spark locations, we could find the sensitivity of spark detection, the rate of the false positive detection, and the accuracy of spark morphology estimation for different conditions.

For comparison of spark morphology estimation done by our program with the results of SparkMaster, we generated a reduced set of synthetic experiments. Due to a large amount of manual processing involved when performing analysis by SparkMaster and the lack of direct connection to the database, the experimental images were generated with the fixed spark amplitude allowing us to process SparkMaster results easier. For each studied condition (SNR and spark amplitude), five images were generated with a total of ∼300 sparks analyzed per each condition.

Experimental procedures and cardiomyocyte isolation

Animals

C57BL/6J Ola Hsd mice were obtained from Envigo RMS B.V. (The Netherlands) at the age of 6–8 weeks. To acclimatize, they were kept for several weeks at our local animal facility with free access to water and food (V1534-000 Rat/mouse maintenance from Ssniff Spezialditen GmbH, Soest, Germany), an ambient temperature of 22–22.8 °C, and a 12:12 h light:dark cycle.

Cardiomyocytes were successfully isolated from five C57BL/six mice (three male and two female) 208–215 days old. They had a body weight of 29 ± 6 g and a tibial length of 2.22 ± 0.07 cm.

All animal procedures were approved by the Estonian National Committee for Ethics in Animal Experimentation (Estonian Ministry of Agriculture, decision from August 29, 2013, number 12).

Isolation of cardiomyocytes

Cardiomyocytes were isolated using a slightly modified version of a method described previously (Branovets et al., 2013). The mice were anesthetized with a mixture of ketamine/dexmedetomidine (150 and 0.5 mg/kg, respectively) and received an injection of 250 U of heparin to prevent blood coagulation. The heart was excised and immediately placed in ice–cold wash solution (see composition below). It was cannulated via the aorta on a Langendorff perfusion system, which was thermostatted so that the outflow temperature was ∼37 °C. The heart was first perfused with a wash solution at a constant pressure of 80 cm H2O. The initial pressure and the initial flow were recorded. After the heart was washed free of blood, the perfusion was switched to a constant flow (initial flow reduced by 0.5 ml/min) with a digestion solution containing 0.25 mg/ml Liberase DL (Roche, Penzberg, Germany). The pressure, which was monitored continuously throughout the protocol, usually increased transiently. When the pressure had decreased back to the initial value, 1.36 mg/ml of dispase II (Roche) was added to the digestion solution. As the pressure decreased further, the flow was carefully increased up to 3 ml/min. The digestion was continued until the pressure had decreased to 30–40% of the initial value and the heart was soft. After perfusion, the ventricles were cut into smaller pieces, transferred to a beaker with digestion solution and incubated further at 37 °C with gentle shaking until the tissue started to dissociate. As a result, the cell suspension was a mix of isolated cardiomyocytes from the left and right ventricles. Cells were harvested with a pasteur pipette several times during this post-digestion and filtered through a 100 μm cell strainer (EASYstrainerTM Cell Strainer; Greiner Bio-One, Kremsmünster, Austria) into a vial with sedimentation solution. Each time, additional digestion solution was added to the undigested cells. The viable cells were separated by sedimentation or by centrifugation for 2 min at 300 rpm/12 g in an Eppendorf 5810R centrifuge with an F-34-6-38 rotor (Eppendorf AG., Hamburg, Germany). First, extracellular Ca2+ was gradually increased to two mM to ensure Ca2+ tolerance of the cells. After this, extracellular Ca2+ was washed out again by rinsing the cells three times with five ml of sedimentation solution. The isolated cells were stored in this solution at room temperature until use.

Solutions

The wash solution consisted of (in mM) 117 NaCl, 5.7 KCl, 1.5 KH2PO4, 4.4 NaHCO3, 1.7 MgCl2, 21 HEPES, 20 taurine, 11.7 glucose and 10 2,3-butanedione monoxime. pH was adjusted to 7.4 with NaOH.

For the digestion solution, 0.25 mg/ml Liberase DL (Roche) and 1.36 mg/ml Dispase II (Roche) were added to 20 ml of the wash solution.

For the sedimentation solution, two mM pyruvate, 10 μM leupeptin (Roche), two μM soybean trypsin inhibitor, and three mg/ml BSA (Roche) were added to 40 ml of the wash solution.

For the microscopy experiments following solution was used (in mM): 150 NaCl, 5.4 KCl, 0.33 NaH2PO4, 1 MgCl2, 1.13 CaCl2, 10 glucose and 10 HEPES. pH was adjusted to 7.4 with NaOH.

All chemicals were obtained from Sigma-Aldrich (St. Louis, MO, USA) if not mentioned otherwise.

Experiments

The experiments were performed by initially pacing cardiomyocytes at one Hz either in the presence (50 nM) or absence of isoprenaline (ISO) for about 5 min at room temperature of 25 °C. A contracting cardiomyocyte, responding to each stimulation pulse and without spontaneous activation, was selected. Before experiment on the selected cardiomyocyte, an image was recorded with the background to obtain background intensity estimate. Then, confocal imaging was started while the cell was still being paced. After a few seconds of imaging, pacing was stopped and the occurrence of sparks was recorded. For each experiment, a linescan image was acquired. After an experiment on one (80% of cases) or two cells, the cells were replaced with a new batch.

Fluorescent labeling was started 15–20 min before experiments, each batch separately labeled just before placing the cells into the measurement chamber under the microscope. For labeling, five μM of Fluo-4 AM (Invitrogen, Carlsbad, CA, USA) was used. One mM Fluo-4 AM stock solution was prepared by diluting 50 μg in DMSO containing 20% (w/v) of Pluronic F-127 (Invitrogen, Carlsbad, CA, USA).

Confocal images were acquired on an inverted confocal microscope Zeiss LSM510 Duo built around Axio Observer Z1 with a 63×water-immersion objective (1.2 NA) at room temperature. The signal was acquired via high-voltage single PMT using 8-bit mode; the pinhole was set to one Airy disk. Line scans were performed unidirectionally. xt images were made to cover a line of ∼80 μm (512 pixels, 0.15 μm pixel size) along the cell during ∼90 s (60,000 lines, 1.52 ms apart). The imaging was performed in RC-21BRFS chamber (Warner Instruments, Hamden, USA) with a coverslip on the bottom only. The chamber was connected with Stimulator C Type 224 (Harvard Apparatus, Holliston, Massachusett, USA) to pace the cells at one Hz using square pulses (width 10 ms, height 20 V). Fluo-4 was excited by Argon laser at 488 nm (about 1% of the power) with the fluorescence collected through a long-pass 505 nm filter after passing 488 nm dichroic.

Statistics

When not stated otherwise, statistical measures are reported as mean ± SD. Statistical tests were performed using JASP (JASP Team, 2018). For Bayes Factor interpretation, common evidence categories were used (Lee & Wagenmakers, 2014). For analysis with JASP, SQL, and Python scripts were written to fetch the data from the database into the file in the format suitable for JASP input.

Results

Results are organized as follows: First, the user interface is described along with the analysis workflow. Second, the spark detection algorithm is characterized, followed by the evaluation of spark morphology estimation by the software. Finally, the analysis of experimental data is shown to illustrate the capabilities of the software. During the evaluation of the software, results are compared with the results obtained by SparkMaster (Picht et al., 2007)—widely used software for Ca2+ sparks analysis.

User interface

The GUI gives an overview of the experiment as well as allows to examine each spark individually (Fig. 1). At the start of the program, a user can either open a new experimental file or choose an experiment that has been analyzed earlier. The whole experiment is expected to be recorded as a single image. This image is shown in the central part of the GUI. It can display large datasets. For example, our studies routinely used 512 × 60,000 (xt) images. Pixel dimensions in space (x) and time (t) coordinates are scaled separately, allowing the user to get an overview of the experiment as well as zoom into a region of interest through scrolling mouse wheel on the overall image or its axes. Average intensity changes in time are shown on the top together with intensity histograms for the whole experiment and the region selected for sparks detection, allowing a quick assessment of the experiment.

The analysis of the experiment starts with the definition of experiment stages. The stages are defined by the time interval and cover the whole image in the spatial coordinate. The first stage is the one that will be analyzed for sparks. The other stages can be defined by the user for the assistance in statistical analysis of the experiment. In our case, we field-stimulated the cells in the first few seconds of the experiment allowing us to follow calcium transients during normal EC coupling. For each user-defined stage, mean, minimal, maximal, and values corresponding to 0.1% and 99.9% percentiles were found. The percentile values are useful to estimate the range of the signal changes without being overly sensitive to one spurious data-point, as minimal and maximal values are.

When analyzing new experiment, the user has to define the experimental stage in which sparks are detected and, under “Image analysis parameters” subsection of “Parameters,” define the background intensity. The background intensity corresponds to image counts recorded in the absence of fluorescence. The background intensity is determined by the detector only and should be measured with the same detector settings, such as gain, pixel time, and offset, as during the experiments in the absence of illumination. In practice, we recorded a separate image before performing the sparks experiment and used it to obtain the background intensity. As soon as the sparks detection experiment stage and background intensity are defined, the user can start the automatic detection process by pressing “Detect.” At first, the image is normalized by finding slowly changing background fluorescence for each point of the cell. The calculated background fluorescence is available under “F0” tab in the image view. Next, the variability of the noise induced by uneven background fluorescence is accounted for with the result shown under “Corrected” image tab, the corrected image blurred (“Medblu” image tab), and, sparks detected (“Filtered” image tab). Example images corresponding to the different stages of the analysis are shown in Fig. 4. As explained in ‘Methods’, due to the even distribution of the noise along the sample in the corrected image (Fig. 4E), detection of the sparks by comparing local deviation of the intensity from the background was insensitive to the spatial unevenness of the mean fluorescence during an experiment.

Figure 4 Different stages of the experiment analysis.

From the original recording (A), a sparks detection experiment stage is selected (B). In that stage, the background is determined (C) allowing to calculate a background subtracted and uneven noise corrected image (D) with its median-blurred version (E). For each detected spark (F), its temporal and spatial properties are determined on the basis of the spline fits, as in (G) and (H), respectively.

With the sparks regions detected, whether earlier and loaded from the database or after automatic detection, properties of each spark are calculated. The user can examine each spark by either selecting it in the full experiment overview by hovering a cursor over it or using a list of sparks on the bottom left. The user can adjust the region for each spark by either moving it or changing the size in the experiment overview. In practice, we are zooming in into the region with the spark, changing the region, and zooming out. The regions can also be removed by the right click of the mouse either on the sparks list or experiment overview. To create a new sparks region, the user can click and hold the right mouse button and select it via dragging the pointer in the experiment overview.

The default parameters for sparks intensity fitting can be provided by the user under “Parameters” in the “Spark analysis parameters” sub-section. Sometimes, one has to adjust the fitting parameters for some individual spark region, due to the longer duration of Ca2+ release, for example. This can be done when selecting the corresponding spark region, changing the corresponding parameter, and pressing Enter on the keyboard to initiate a new fit.

When analyzing an individual spark by selecting its region, the corresponding original image, corrected, and blurred corrected images at the bottom left of the GUI are shown. These images also show the detected spark surrounded by red contour and the main axes used to determine spatial and temporal characteristics of the spark. Below, the temporal and spatial variation of intensity along the principal axes is shown together with the spline fits allowing the user to access the quality of the fit with the current fit parameters (see, e.g., Fig. 4). The found properties of the spark are shown on the right of the individual spark images and stored, together with the fit parameters and region definition in the database.

The results of the analysis are stored in the database, as described in ‘Methods’.

Evaluation of the spark detection algorithm

Spark detection algorithm is based on threshold approach by Cheng et al. (1999) and has similar characteristics to the other implementation of this algorithm (Picht et al., 2007). In contrast to earlier evaluations (Picht et al., 2007), as described in ‘Methods’, we generated synthetic data assuming the Poisson distribution of the noise, as expected in the confocal microscope if the digital offset is taken into account.

In the example simulation in Fig. 5, sparks with different amplitudes were detected perfectly. Detection success rate depends on the spark amplitude and the noise level, as shown in Fig. 6A. As expected from the earlier analysis of similar algorithms (Cheng et al., 1999; Picht et al., 2007), the sensitivity of spark detection increases with the increase of the amplitude and the reduction of the noise. At used spark detection parameter values (see ‘Methods’), a number of false positive spark detections were below 0.1 sparks per second per 100 μm, at all analyzed noise levels.

Figure 5 Example of synthetic dataset used for evaluation of the spark detection algorithm.

Image with randomly distributed sparks with amplitudes of 0.5, 0.75, 1.0, 1.25 × F0 (A) was disturbed by Poisson noise to obtain SNR of the background equal to 2 (B), and analyzed by sparks detection algorithm using smoothed image (C). Notice how all the sparks were detected, as indicated by the white boxes on (C).

Figure 6 Analysis of the performance of the detection algorithm.

In (A), the sensitivity of the spark detection strongly depends on the SNR of the image and the amplitude of the detected spark. Here, detection sensitivity for a given amplitude was defined as a proportion of detected sparks out of all sparks with that amplitude. In (B), false positive events characterized as sparks (false positive sparks) are shown as a frequency of the events. The frequency for false positive sparks with given amplitude As was defined as frequency of events with the amplitudes that were at least As. For example, for SNR 2, the false positive sparks had an amplitude which was smaller than 0.7 × F0 and larger than 0.4 × F0. As with sensitivity, there is a large influence of SNR and amplitude on the false positive sparks frequency. In (C), positive predictive value (PPV) is shown against the spark amplitude estimated by the program. Here, PPV was found by binning results with a step 0.05 × F0 and only conditions with more than 20 events (true or false positives) are shown. See the main text for interpretation of the results.

Similar to the effect observed in Picht et al. (2007), there is a higher false positive spark detection rate at low noise images at certain conditions. To analyze the spark frequency statistically, we calculated, for every given cut-off amplitude, the frequency of the false positive sparks with the amplitude larger than the cut-off amplitude. For example, for the cut-off amplitude of 0.5 × F0, we found the frequency of false positive sparks detected by the algorithm with the amplitude larger than 0.5 × F0. Our analysis demonstrates that the false positive sparks amplitude depends on the image noise, with the amplitudes ranging from 0.2 × F0 (SNR = 4) to 1 × F0 (SNR = 1.4).

In our experiments, after subtraction of the digital offset, SNR of the background was about 2. According to Fig. 6B, in these conditions, we expect the detection of false positive sparks with the amplitudes mostly from 0.5 × F0 to 0.6 × F0. Note that between these amplitudes, the cumulative frequency of sparks below given amplitude changes the most for simulations with SNR equal to 2 (Fig. 6B). The false positive sparks detection frequency was below 0.07 spark/(s · 100 μm). As for sensitivity, all sparks with the amplitudes higher than 0.4 × F0 are expected to be detected (Fig. 6A) with the 50% detection success at the spark amplitudes of about 0.3 × F0.

In practice, we do not have a knowledge regarding the true spark locations nor their amplitudes, but we have to extract this information from the estimation done by the spark detection program. For judging the performance of the algorithm using the estimated spark amplitudes, we found positive predictive value (PPV; contribution of true positives to total count of detected events, PPV) and related it to the estimated spark amplitude (Fig. 6C). This is in contrast to Fig. 6A, where the true spark amplitude was used to estimate the sensitivity. As shown in Fig. 6C, PPV values are close 100% for the most of the studied conditions at the spark amplitudes above 0.8 × F0. The exception is for SNR = 1.4, where 100% is approached toward spark amplitudes of 1 × F0.

We compared the performance of our software with SparkMaster’s (Picht et al., 2007) analysis on the same dataset, the experiment shown in Fig. 4. While our software detected 132 sparks in selected region of interest, SparkMaster detected 219 sparks. Among the detected sparks, 122 sparks were detected by both programs, with a strong correlation of the spark amplitudes found by both programs (r2 = 0.82). Most of the sparks found by only one of the programs were with the small amplitude, as shown in Fig. 7. As illustrated in Fig. 7, SparkMaster tends to find a much larger number of smaller sparks than our software.

Figure 7 Comparison of spark detection by our program and SparkMaster using the data from Fig. 4.

The detected sparks were grouped into three groups: sparks detected by both programs (“Both”), detected by our software (“IOCBIO Sparks”) or SparkMaster (“SparkMaster”) only. Spark amplitudes are normalized by SparkMaster and our software differently due to the handling of F0 (see Discussion). To correct for the differences in amplitudes normalization, the amplitudes found by SparkMaster were mapped to the amplitudes corresponding to our software through a linear relationship between the amplitudes. Note that the sparks detected with the both programs have relatively large amplitudes and, in contrast, the sparks detected by only one of the programs are mainly restricted to the sparks with small amplitudes.

To test whether large number of smaller sparks found by SparkMaster can be explained by false positives, we analyzed synthetic images with SNR = 2 using SparkMaster. In these tests, we observed a significantly larger number of false positive spark detections by SparkMaster than with our algorithm. SparkMaster got a false positive rate of 0.36 ± 0.08 spark/(s · 100 μm), mean ± sem (n = 5). For comparison, our algorithm at the same conditions, had a false positives detection rate below 0.07 spark/(s · 100 μm) (Fig. 6B). Thus, in part, the difference in spark frequency estimation between the programs can be explained by larger false positive spark detections by SparkMaster. However, note that this difference was mainly observed for smaller sparks and the spark detection was matched closely for the larger sparks by the both programs.

Evaluation of the spark morphology estimation

Using the same synthetic data as for evaluation of spark detection algorithm, we looked into estimated spark amplitude and its temporal (FDHM) and spatial (FWHM) characteristics. As shown in Fig. 8, estimation of spark morphology depends on SNR and the spark amplitude. At higher amplitudes and lower noise levels, spark amplitude, FDHM, and FWHM are perfectly determined.

Figure 8 Analysis of the performance of the spark morphology estimation.

For all successfully detected sparks from the synthetic data, mean estimated amplitude (A), FDHM (B), and FWHM (C) were related to the ground truth values that were used to generate synthetic data. The values used to create synthetic spark data are indicated in the plots using a dashed line (Expected). In the analysis, we varied SNR and spark amplitude with the fixed FDHM and FWHM used to generate synthetic spark images. Data were analyzed either by our program or SparkMaster, see legend for the used notation. Here, only conditions with at least 25 detected sparks were analyzed. Note how the mean estimates depend on SNR and spark amplitude with the similarity in the estimates done by both programs.

With the increase of noise level, the estimated spark amplitude is somewhat higher than the “true” value. This is a side effect induced by the freedom of the algorithm to select the maximum from detected spark region as a spark amplitude. Indeed, in the noisy image, one can imagine that there could be regional increases in fluorescence in the neighborhood of the spark maximum. If such region exists, it will be selected as a spark maximum and the amplitude will be estimated using it. As a result, a bias towards higher amplitudes is an expected outcome of the algorithm and plays a bigger role with the increase of the noise.

Full width at half maximum and FDHM follow the trends that are similar for the both of them. Estimation of these parameters has a better precision for sparks with the larger amplitudes and measured in the conditions of lower noise. Increase of the noise and reduction of the amplitude lead to underestimation of the spatial and temporal spread of the spark.

For comparison, we analyzed a smaller set of synthetic data by SparkMaster. The results are shown in Fig. 8 for the selected amplitude and noise levels. As it is clear from the comparison between the results obtained by SparkMaster and our software, estimates of amplitude, spatial, and temporal characteristics follow the same trend as the estimates by our software. Interestingly, SparkMaster underestimates spark amplitude at low noise and high amplitude case (SNR 4 and the relative amplitude 2 × F0). Whether it is due to overestimation of the image background or some other reason is not clear to us.

Effect of isoprenaline on sparks

To illustrate the use of the software, we analyzed the effect of ISO in the solution surrounding the cardiomyocytes. As expected, in the presence of ISO, the calcium transient during external pacing was much larger, as evidenced by changes in Fluo-4 labeled fluorescence during each beat. While in control, the amplitude of fluorescence change normalized to F0 was 2.6 ± 0.9 (n = 16 cells), in the presence of ISO, the amplitude was significantly larger at 7.4 ± 1.8 (n = 20, comparison done by Welch test, p < 0.001). However, note that F0 could correspond to the different basal calcium values in control when compared with the measurements in the presence of ISO. So, one should be aware of this normalization while making the comparisons.

As shown in the representative experiments (Fig. 9), sparks are more frequent in the presence of ISO when compared to control. To analyze the spark frequency statistically, we calculated, for every given cut-off amplitude, the frequency of sparks with the amplitude larger than the cut-off amplitude. For example, for the cut-off amplitude of 1 × F0, we found the frequency of sparks with the amplitude larger than 1 × F0. As shown in Fig. 10A, the presence of ISO leads to a higher frequency for sparks with the larger amplitudes than in control. The obtained spark frequencies were significantly higher than the frequency of expected false positive spark detection (Fig. 6B), with the frequency of detected sparks exceeding the one expected for false positives by 4.5× in control and more than 15× in the presence of ISO at all amplitudes. When the frequencies corresponding to the cut-off F0-normalized amplitudes of 0.5, 0.75, 1.0, 1.25, and 1.5 were compared using classical repeated measures ANOVA, the difference was statistically significant (p < 0.001). Very strong evidence for the statistical difference between control and ISO cases was confirmed by Bayesian repeated measures ANOVA, with Bayes Factor BF10 > 70, very strongly suggesting effects of ISO (BF10 > 70) and its interaction with the cut-off amplitude (BF10 > 100). For comparison, the sparks detected by SparkMaster were analyzed in the same manner and we observed strong effect of ISO on increasing the spark frequency (repeated measures ANOVA, p = 0.02) and interaction between ISO and the cut-off amplitude (Bayesian repeated measures ANOVA, BF10 > 100).

Figure 9 Representative experiments showing the effect of isoprenaline as studied by Fluo-4 fluorescence confocal images.

The overall intensity of Fluo-4 fluorescence is significantly higher in the presence of isoprenaline during the pacing stage of experiment (A). The original recordings (B, C) and the corrected images (D, E) show a larger spark frequency in the presence of isoprenaline (B–E have the same scale). We also observed the occurrence of longer calcium release events (F) and multiple sparks occurring at the same location in the experiments with isoprenaline (G).

Figure 10 Analysis of spark frequency for different spark amplitudes.

In all graphs, the frequency was found for occurrence of sparks with the amplitude that was equal or larger than the one indicated on x-axis. The frequency is shown by its mean value (solid line) and the area surrounding it as ± SEM. In (A), the frequency for all sparks is shown in control and in the presence of ISO. In (B), the frequency of longer sparks (sparks with full duration at the half maximum of at least 25 ms) is shown. In (C), the frequency of sparks followed by a spark of at least the same cut-off amplitude within 2 s and within one μm. In (D), similar to (C), but with the two sparks in the location within one μm of the original spark and each spark following the previous one within 2 s. Note that the presence of ISO increased overall spark frequency, spark frequency for sparks with the larger normalized amplitude, spark frequency of longer sparks, and occurrence of multiple sparks in the vicinity of each other at the short period of time.

The sparks with longer duration were analyzed by comparing the frequencies at different cut-off amplitudes. Here, we looked at the sparks with the FDHM that was larger than 25 ms. A total of 25 ms was selected as a representative longer spark duration, as indicated by the analysis of FDHM for larger sparks (described below). As shown in Fig. 10B, the presence of ISO increases the spark frequency for longer sparks (classical repeated measures ANOVA for the same amplitudes as above, p < 0.005). The strong evidence for the difference between ISO and control cases (BF10 > 100) was attributed to the effect of ISO (BF10 > 15) and interaction between amplitude and ISO treatment (BF10 > 100).

As we observed sparks at the same location occurring close to each other in the presence of ISO, we analyzed the frequency of spark groups consisting of at least two or three sparks within given temporal and spatial limitations: less than 2 s between consecutive sparks and within one μm of the location of the first spark. The frequencies were higher in the presence of ISO as shown in Figs. 10C and 10D for two- and three-spark groups, respectively. The difference between frequencies at the used cut-off amplitudes measured in control and ISO were statistically significant for two-spark (classical repeated measures ANOVA, p < 0.05) and had a tendency toward larger frequencies in occurrence of three-spark groups in the presence of ISO when compared to control (p = 0.1). When analyzed using Bayesian repeated measures ANOVA, the very strong evidence for the difference between frequencies of two-spark groups was found, with BF10 > 100, attributed to the interaction between amplitude and ISO treatment (BF10 > 100). Thus, our analysis confirmed the higher chance of occurrence of multiple sparks in the vicinity of the same location.

From the analysis of the overall distribution of spark amplitudes (Fig. 11A), it is clear that the presence of ISO leads to the larger spread among amplitudes of the sparks. Due to the noise in experimental recordings and spark detection algorithm, the amplitude distribution has a cut-off at lower amplitudes, consistent with the analysis of spark amplitude distributions (Izu, Gil Wier & William Balke, 1998; Cheng et al., 1999). Consistent with the changes in the amplitude distribution, the average spark amplitude was larger in the presence of ISO (1.16 ± 0.46, n = 1,063) than in control (0.83 ± 0.30, n = 302), as tested by Welch test (p < 0.001). A same statistically significant increase in average spark amplitude was observed when the sparks detected by SparkMaster were analyzed (Welch test, p < 0.001).

Figure 11 Analysis of spark distribution shown through probability density functions.

(A) Distribution of spark amplitudes in control and the presence of ISO. Notice the shift of the probability density function towards larger amplitudes in the presence of ISO. (B) Distribution of full duration at half maximum for sparks with the amplitudes in the range from 0.8 × F0 to 1.2 × F0. (C) Distribution of full width at half maximum for sparks with the amplitudes in the range from 0.8 × F0 to 1.2 × F0. Note that the temporal and spatial properties for the sparks with the same amplitude were similar in control and the presence of ISO.

While we found that the presence of ISO increased the spark frequency, amplitude, and occurrence of longer sparks as well as the sparking activity from the same site, we characterized whether most of the sparks also had some differences in duration or spark width induced by ISO. For that, we selected the sparks with the amplitudes within the range that would have a significant number of sparks in control and ISO cases: 0.8 × F0 to 1.2 × F0. When comparing the probability densities for FDHM and FWHM between sparks detected in control and in the presence of ISO, we observed that the probability densities were very similar (Figs. 11B and 11C). For statistical analysis, we binned the sparks into two groups according to the amplitude: 0.8 − 1.0 × F0 and 1.0 − 1.2 × F0. The mean FDHM and FWHM were not statistically significantly different between control and ISO cases when analyzed with the Bayesian ANOVA when the spark amplitude binning was included into the null model. A same conclusion was reached when analyzing sparks morphology estimated by SparkMaster. Namely, with the same statistical approach as used above, no statistically significant difference was found for FDHM and FWHM of sparks recorded in the presence or absence of ISO. Thus, we conclude that the duration and the width of the sparks, on average, was not influenced significantly by the presence of ISO.

Discussion

We have developed software for detection and analysis of Ca2+ sparks in cardiomyocytes from line scan confocal images. The software is released under an open-source license allowing free usage and open development of the software. The software is specially developed for spark analysis and has a GUI interface that is designed to make the analysis simple and informative. The results of the primary analysis are stored in the SQL database, allowing integration with large data analysis tools, export to the spreadsheets, and other means of cooperation within and between research groups. We analyzed the sparks detection algorithm and found that sparks were detected reliably under a wide range of conditions with the spark morphology estimated well for larger sparks. In comparison with SparkMaster (Picht et al., 2007), the both algorithms had similar spark morphology estimation properties but our software detected far less false positives (about 5× less in the tested condition). As an illustration of the software capabilities, we demonstrate an increase in spark frequency, tendency of having larger spark amplitudes, sparks with long duration, and occurrence of multiple sparks from the same site in the presence of ISO. We show that the duration and the width of sparks with the same amplitude were similar in the absence and presence of ISO.

Validation of the spark detection and morphology estimation

The verification results clearly show the ranges where sparks are detected reliably and the expected false spark detection frequency. For verification of the spark detection algorithm, we generated a large number of synthetic images that were degraded by Poisson noise. The analysis results (Fig. 6) were similar to the earlier analyses by others (Cheng et al., 1999; Picht et al., 2007), although in those studies Gaussian noise was used to degrade the synthetic images. Note that underestimation of detected spark frequency at low amplitudes is an expected outcome of the algorithm, as analyzed in detail by Cheng et al. (1999).

From the analysis of spark morphology estimation, in terms of the determination of spark spatial and temporal characteristics, we found that the estimates were correct for larger sparks or lower noise cases. The general properties of the morphology estimation were the same as for SparkMaster (Fig. 8). We have limited the analysis to the most robust spark characteristics, amplitude, FWHM, and FDHM. These parameters do not depend on the determination of the spark peak location in time and space, which is usually difficult to estimate in noisy signals without any specific model of the event, the model that we avoided in the current implementation.

The validation results (Figs. 6 and 8) give clear guidelines to the use of the implemented algorithm in practice. In particular, to avoid the influence of the false positives, it is advisable to limit the statistical analysis to sparks with the range of amplitudes at which the number of the false positives is significantly smaller than the recorded one. In addition, when analyzing spark morphology, larger sparks should be preferred. Specific ranges would have to be determined for each experiment series separately from the comparison of the experimentally observed SNR and the analysis results of the algorithm (Figs. 6 and 8). However, to make such an informed decision, the users do need to have such validation results, as provided in this work, and not assume that a used spark detection algorithm works perfectly in any condition.

Comparison with the other spark detection algorithms

The presented algorithm is based on thresholds approach by Cheng et al. (1999), Picht et al. (2007) and has a similar performance, as it is clear from the analysis of synthetic datasets. There are, however, some important differences.

When analyzing experimental data, we fit the background of the dataset using a spline with the set nodal points, one spline fit per each spatial location. This allows taking into account slow variation of the background. In contrast, Cheng et al. (1999) used a single average line for background estimation which was extended up to five intervals by Picht et al. (2007). As a result of this difference in the background estimation, our software was able to analyze longer experiments performed as a single acquisition (60 s and more) than with SparkMaster. In particular, SparkMaster was struggling to take into account changes in background occurring in longer experiments.

In Cheng et al. (1999), during spark detection, a median filter was used on binary images corresponding to the peak and overall spark intensity regions. We used the same filter for the peak intensity and applied minimal spark size requirement. As it is clear from the synthetic data analysis, this performed well in terms of reducing the amount of false positive detections. We could not confirm or deny whether SparkMaster is using median filters in binary images, as in (Cheng et al., 1999), due to the inability to access the source code. In addition, in the corresponding paper (Picht et al., 2007) the usage of median filter was not mentioned, the omission of this filter was not stated either.

From the comparison of the performance of our software with the results of SparkMaster using the same experimental data, we found that sparks with larger amplitudes are equally well detected with both software (shown in Fig. 7). The differences in detection were observed for smaller sparks, which may possibly be explained by a larger false positive detection rate with SparkMaster. However, over a certain threshold, the same sparks were detected by both programs, as expected for programs using the similar algorithms.

In our analysis, we subtract the offset corresponding to the dark current recordings, as supplied by the user. As a result, the sparks are expected to be detected for all correctly used offset values and will not depend on whether the digital offset was set higher or lower during the recordings, as long as the data is within the digitalization range of the image. This is in contrast to (Picht et al., 2007), where F0 is calculated with the digital offset as a part of F0. By subtracting the dark current readout, we can normalize fluorescence readouts taking into account the Poisson nature of the noise and detect sparks using a deviation from this distribution, a similar approach to the variance stabilization used in Bankhead et al. (2011). This allows to avoid false positives induced by larger standard deviation of background signal in the regions with higher fluorescence, as is expected in the algorithms that use the same standard deviation for the full data range.

The speed of data analysis using our software was either similar to SparkMaster or slower, as determined on the smaller datasets that SparkMaster was able to analyze somewhat faster before hitting software limitations induced by a small number of background estimation intervals or a maximal number of sparks (up to 500 sparks per image). The performance of our software depended strongly on used spark profile fitting splines, with the monotonic splines, while fitting data better, were considerably slower than non-monotonic ones.

Software design

The software was designed to assist researchers in the analysis of sparks recording experiments. All the steps required for the primary analysis of the data are easily available. For control over the overall quality of the experiment, the full experiment overview is presented to quickly determine such aspects as a movement of the cell during an experiment. As a part of the overall experiment quality assessment, the displayed intensity profile allows to check whether the overall calcium concentration was stable during an experiment, and, through shown histograms, whether imaging parameters were selected correctly. The experiment analysis is streamlined and has been partitioned into overall image normalization, spark detection, and spark analysis stages. Therefore, the researcher has full control over each of the analysis steps ensuring the image is appropriately normalized, sparks are detected, and, for each spark, the temporal and spatial fits have been obtained.

By developing a dedicated program, we were not restricted in a selection of user interface elements and interaction between them. As a result, we could link overview of the experiment, spark regions in the overview images, and spark analysis data, allowing the researcher to assess the spark measurements in isolation or together with the surroundings. This all should assist the analysis and, as a result, lead to a better quality of the results.

Handling of the analysis data

As an analysis results storage media, we selected SQL database. Through SQL database, it is easy to access and analyze the data as well as export it into spreadsheet format, if needed. At present, there are two databases supported: PostgreSQL and SQLite. For the research teams that are not routinely using databases for data analysis, SQLite database does not require any installation and will be created by the program on the first run. Data can be shared then either in the form of the database file or through spreadsheets. When the central PostgreSQL database is available, the data analysis results can be directly stored into it. Such storage in the central database, combined with the central storage of experiment images, has many advantages and we would like to recommend this approach to others. In particular, by storing the data centrally, the experiments and the analysis is accessible to everyone in the research team, as set by the database permissions. Since each experiment and each spark is assigned a unique ID, the spark analysis results can be related to all other experiments done on the same cardiomyocyte preparation, or animal phenotype, or any other aspect, such as details of cardiomyocyte isolation. With the development of the large set of data exploration tools, the analysis of such relationships could lead to discoveries on the basis of recorded data.

Open source

The software was released together with its source code under an open-source license—version 3 of the GNU General Public License. This license clearly defines the usage of the software as well as its further development. The license has been selected to allow interested research teams to develop the software further and to be ensured that their work will be kept as open. In addition to the open-source license for our software, we selected only open-source software as a requirement for it. This allows using the software without any external environment with the limited availability, such as MATLAB or IDL, platforms of choice for some other spark analysis programs (Davoodi et al., 2017; Cheng et al., 1999; Kong, Soeller & Cannell, 2008). As a result of the usage of only open-source components, it makes it easy to test, examine the algorithm details, and improve the developed software further by any interested researcher. For example, such algorithms for sparks detection as in (Kan, Yip & Yang, 2015) can be incorporated into the software.

The software was developed in an open manner with the code released through the version control repository management system. This makes all changes in the code documented and traceable, an important requirement in ensuring reproducibility of the research results obtained with this software.

Software limitations

There are some limitations of the current version of the software, and the most important ones are addressed below.

Sometimes the same spark is detected multiple times. This happens when a smaller spark is detected next to the larger one with the corresponding regions of the secondary analysis overlapping each other. This is a drawback of the used algorithm for the secondary analysis of the detected spark and can be handled either by the user through correction of the region of analysis for the smaller spark or removing the region of analysis. The latter approach would reduce the number of detected sparks, but, taking into account the relatively high frequency of smaller sparks, should not significantly impact the data analysis results.

The algorithm was developed and tested for spark detection in the part of the recordings were the cells were unpaced. To extend it to the paced cells, we expect that the background subtraction algorithm would have to be enhanced. At present, with the fast changes in overall calcium-induced fluorescence during pacing, the estimated background tends to fit the data relatively poorly, probably due to the spline used for the background estimation.

While the cell is not paced, it is possible that Ca2+ waves occur during the experiments. Considering that the algorithm is developed to take into account a slowly changing background, the same limits, as mentioned above regarding pacing phase, are imposed. However, it is possible to handle occasional Ca2+ waves in the experiments by using relatively large interval between nodes of the spline approximating the image background. In this case, recordings during Ca2+ wave will be not used for the background estimation and will be detected as a large spark or ignored if the wave extends to at least one of the edges of the recordings. Since the software includes an opportunity to discard the incorrectly determined sparks, the area corresponding to the wave can be removed when reviewing the experimental analysis. Note that using a spline with a small interval between nodal points (with the hope of removing Ca2+ wave induced fluorescence as a background) frequently leads to the situation where the background estimation fails due to the used spline. For reference, we use 5–10 s interval between nodal points in these cases.

At present, all calculations are done through a single working thread. This is a current limitation that is enforced by the packages used to develop the software. However, the software architecture allows overcoming these limitations through separation of calculation and user interface parts by partitioning currently single-program into separate processes and utilizing one of the intra-process messaging libraries to communicate between the parts. Such development is planned for the future versions of the software and should speed up calculations notably.

As an alternative to implementing parallelism in the analysis of a single experiment, one can also run multiple instances of the analysis software to analyze multiple experiments in parallel. This requires that the analysis data is stored in the database supporting parallel access well, with PostgreSQL recommended for such usage. In our practice, we have been analyzing about 6–10 experiments simultaneously which made the analysis time-limited from analysis speed by software to inserting and reviewing of the data by a researcher.

As an advantage of the used open-source software license, the software limitations can be also addressed by community members and not rely only on one research team to resolve them all. We expect that such an open model would also allow using the developed software for testing other approaches for calcium sparks detection that would advance the field further.

Study limitations

We have tested the performance of our software with the experiments done on mouse cardiomyocytes. This is a model that is established in our laboratory and, in agreement with the replacement, reduction, and refinement of animal experiments, we supplemented the tests against experimental data with the tests on synthetic images. Since the performance of software and the test results were similar to the results obtained by others (Cheng et al., 1999; Picht et al., 2007), we expect that the spark detection algorithm will perform for other types of cells as well. While it may require some tuning of the parameters, there are no limitations by the algorithm expected when compared to the results obtained with SparkMaster (Picht et al., 2007), for example. However, it has to be verified by others using different types of cells.

Comparison with earlier results

As an example test case, we looked into effect of β-adrenergic stimulation on the sparks. In agreement with others (Santiago, Ríos & Shannon, 2013; Zhou et al., 2009; Fernández-Velasco et al., 2009), we have found an increase in spark frequency in the presence of ISO (Fig. 10A). The sparks frequency in control was also similar to the frequency reported by Ferrier, Smith & Howlett (2003) at a similar temperature. Moreover, we observed significantly more sparks detected from the same site in the presence of ISO (Figs. 10C and 10D), in agreement with others (Ramay, Liu & Sobie, 2011; Santiago, Ríos & Shannon, 2013). When looking into spark properties, on average, in β-adrenergic stimulated cells, sparks have larger amplitudes than in control, as in Zhou et al. (2009). So, as demonstrated in this example case, the analysis performed by our software reproduces the results of others.

Conclusions

In summary, we have developed and released an open-source calcium sparks detection program. The program was designed to streamline the primary analysis of experiments and, by the usage of the database backend, to the follow-up analysis of the data. Through the use of only open-source components, the usage of the program and its further development does not require any additional commercial licenses. By targeting only calcium spark detection and through open sourcing our work, we hope that many research teams in the field will find it useful.

Supplemental Information

Supplemental Information 1 Detected spark without ISO.

Sparks and their characteristics detected for experiments performed in the control conditions.

Click here for additional data file.

Supplemental Information 2 Detected sparks, ISO.

Sparks and their characteristics detected for experiments performed in the presence of isoprenaline.

Click here for additional data file.

Additional Information and Declarations

Competing Interests

Author Contributions

Animal Ethics

Data Availability

The authors declare that they have no competing interests.

Martin Laasmaa conceived and designed the experiments, contributed reagents/materials/analysis tools, prepared figures and/or tables, authored or reviewed drafts of the paper, approved the final draft, wrote software.

Niina Karro contributed reagents/materials/analysis tools, approved the final draft.

Rikke Birkedal contributed reagents/materials/analysis tools, authored or reviewed drafts of the paper, approved the final draft.

Marko Vendelin conceived and designed the experiments, performed the experiments, analyzed the data, prepared figures and/or tables, authored or reviewed drafts of the paper, approved the final draft, wrote software.

The following information was supplied relating to ethical approvals (i.e., approving body and any reference numbers):

All animal procedures were approved by the Estonian National Committee for Ethics in Animal Experimentation (Estonian Ministry of Agriculture; Decision from August 29, 2013, number 12).

The following information was supplied regarding data availability:

The code is available at GitHub: https://gitlab.com/iocbio/sparks. The original raw data is available as Supplementary Files.

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
