# Peer review of "IOCBIO Sparks detection and analysis software"

_PeerJ, doi:10.7717/peerj.6652_

## Round 0.1 · original submission · Minor Revisions

Your manuscript has now been seen by 2 reviewers. You will see from their comments below that while they find your work of interest, some important points are raised. After careful consideration, we feel that your manuscript will likely be suitable for publication if it is revised to address these points below. We therefore invite you to revise and resubmit your manuscript, taking into account the points raised. Please highlight all changes in the manuscript text file.

Reviewer 1 ·

Basic reporting

The authors present a new open-source software toolbox for detecting and analyzing calcium release events (called sparks) in confocal images of cardiomyocytes. Calcium sparks provide insights into the calcium handling within the cell which in turn informs the excitation-contraction dynamics in the cardiac cell. The software is claimed to improve that process because it is specialized to detect calcium sparks only.

The main finding is the detection rates of the proposed algorithm (Figure 5) and the comparison to that of the competing SparkMaster software (Figure 8). I found both of those figures rather uncoventional and hence somewhat difficult to interpret. I felt that data might be better shown as conventional Receiver-Operating-Characteristic (ROC) curves.

Otherwise, the paper seems to be technically sound although I felt that it would benefit from some reorganising to bring the GUI (Figure 6) and the detection rates (Figures 5 & 8) forward. Doing so would give the reader an early insight into main findings before getting into the details of the methods.

Experimental design

The principal outcome is software so experimental design is not really an issue.  As mentioned above, I suspect that ROC curves might be a better way to present the detection-rate data.

The experimental preparation of the mice cardiomyocytes is outside of my expertise, so I make no comment on those.

Validity of the findings

No comment

Additional comments

Minor points

line 50: I suggest making your GUI screenshot (Fig 6) the first Figure in your paper. You could describe it here at line 50. That would give the reader an immediate feel for your software before getting into the details of the algorithms.

line 52: Better wording might be: "Lastly, the output is saved to an SQL database for inter-operability with third-party applications."

Figure 3: I think you can omit Figure 3. The reader is unlikely to care about the details of the database schema.

line 100: "used animals" -> "animals used" or "animal data"

line 193: grammar: "statistical measures are reported as"

Figure 4: This is a nice Figure. 

line 202-203: I did not understand this sentence. "As a result, the signal-to-noise ratio (SNR) of the background (here, mean value divided by standard deviation is used as SNR) is defined by its mean value."

Figure 5B: The caption is confusing. Can you please clarify. I suspect that a Receiver-operating-characteristic (ROC) curve might be more appropriate.

line 224: Suggest rewording to " The time series data for the entire recording is shown in the central of the GUI. It can display large datasets. For example, our studies routinely used 512 x 60000 samples."

Discussion: I suggest rewriting to recapitulate what you FOUND rather than what you DID. For example (lines 348-350) We analyzed the sparks detection algorithm and found that sparks were detected reliably under xxxx conditions. In comparison with SparkMaster, both algorithms detected the same true positive but our software detected far less false positives (perhaps even quantify this with some numbers).

Reviewer 2 ·

Basic reporting

Laasmaa et al. are proposing an interesting new software for analyzing calcium sparks which fulfill a lack of open source automated software specifically designed for Ca2+ sparks detection/analysis. This new software has many improvements in comparison with the most used Sparkmaster plugin, and it is more versatile than the software actually available. I found particularly interesting, inter alia, that the user can examine each spark individually and decide to include new ones of to exclude/eliminate not desired false sparks for example. It also allows data storage/handling, and a friendlier GUI than other spark detection software available.
While the proposed software by the authors emerges as an interesting alternative to the actual widely used software, I have the following comments on the manuscript.

Minor comments:
1. Line 7 of first paragraph of introduction: Ca2+ sparks are defined as the fluorescent event detected by confocal microscopy, and not the SR Ca2+ release itself. So, when defining Ca2+ sparks in line 30, use “Ca2+ sparks” after "change in fluorescence of a Ca2+ sensitive dye”.
2. Delete double “the” in Line 339
3. Delete double “the” in Lines 394-395

Experimental design

When opening a file, the software estimates the background image F0 by excluding autodetected sparks. But if the x,t image contains not only sparks but one or more calcium waves, the F0 estimation will generate an error in Fo estimation. Although something related with this is described under Software limitations (paced cells), please include instructions for the user, or discuss the situation in which in the same image coexist Ca2+ sparks and Ca2+ waves, or even with an isolated spontaneous calcium transient.

Validity of the findings

When comparing Iocbio to Sparkmaster, the authors only compare detected number of Ca2+ sparks, Ca2+ spark amplitude and false positive rate. It would be useful to compare the Ca2+ spark morphology results (e.g. FDHM, FWHM, and so on) obtained with the author´s software in the presence/absence of Isoprenaline, with the same images analyzed with Sparkmaster. This will help to validate the novel software.

Additional comments

N/A

---

## Round 0.2 · accepted · Accept

Thank you for the detailed response letter. We are delighted to accept your manuscript for publication.

#